# VISION LANGUAGE MODELS CANNOT PLAN, BUT CAN THEY FORMALIZE?

## ABSTRACT

The advancement of vision language models (VLMs) has empowered embodied agents to accomplish simple multimodal planning tasks, but not long-horizon ones requiring long sequences of actions. In text-only simulations, long-horizon planning has seen significant improvement brought by repositioning the role of LLMs. Instead of directly generating action sequences, LLMs translate the planning domain and problem into a formal planning language like the Planning Domain Definition Language (PDDL), which can call a formal solver to derive the plan in a verifiable manner. In multimodal environments, research on VLM-AS-FORMALIZER remains scarce, usually involving gross simplifications such as predefined object vocabulary or overly similar few-shot examples. In this work, we present a suite of five VLM-AS-FORMALIZER pipelines that tackle one-shot, open-vocabulary, and multimodal PDDL formalization. We evaluate those on an existing benchmark while presenting another two that for the first time account for planning with authentic, multi-view, and low-quality images. We conclude that VLM-AS-FORMALIZER greatly outperforms end-to-end plan generation. We reveal the bottleneck to be vision rather than language, as VLMs often fail to capture an exhaustive set of necessary object relations. While generating intermediate, textual representations such as captions or scene graphs partially compensate for the performance, their inconsistent gain leaves headroom for future research directions on multimodal planning formalization.

## 1 INTRODUCTION

Embodied planning has seen impressive advances in the last few years, especially with the rise of vision language models (VLM) and vision language action models (VLA). The standard setup involves giving the model interleaved vision-language inputs, such as images or videos and a natural language instruction, and expecting the model to predict a sequence of actions that logically form a plan to achieve the specified goal. While either VLM or VLA models may directly output high-level or low-level actions, their performance is limited in long-horizontal planning with little to no interpretability (Liu et al., 2023; Yang et al., 2025b). A mainstream alternative is a hierarchical pipelining approach that involves multiple models for detecting objects, extracting relations, predicting task-level actions, and translating to motion-level actions (Yenamandra et al., 2023; Wang et al., 2024a). While modular, such approach requires training for each modules in specific domains, thus lacking few-shot generalization abilities in an open-vocabulary setting.

Large language models (LLM), the basis of VLM and VLA, have similarly driven great progress in textual planning with two leading paradigms under few-shot settings. Given a textual description of the environment and the goal, LLM-as-planner directly generates the actions (Wei et al., 2025). In addition to mixed performance, this approach offers little interpretability and verifiability. Alternatively, LLM-as-formalizer instead generates a formal language like the Planning Domain Definition Language (PDDL) which can be input into a symbolic solver to derive a plan deterministically (Tantakoun et al., 2025). Powered by pre-trained LLM's strong in-context learning skills and code generation ability, this approach has shown promising performance while offering some formal guarantee that is crucial for high-stakes domains. Despite the emerging success of LLM-as-formalizer, application to VLM has been understudied in multimodal planning environments, with some incomplete attempts leveraging non-visual cues (Li et al., 2025; Kwon et al., 2025) or close-

vocabulary, few-shot examples (Herzog et al., 2025b). As a result, such explorations are limited to particular tasks and lack generality (Jenamani et al., 2025).

We advance VLM-AS-FORMALIZER as an effective paradigm on long-horizon, one-shot, open-vocabulary, visual-language planning tasks. We are the first to systematically evaluate its strengths and weaknesses. To do so, we consider 2 VLMs and design 5 VLM-AS-FORMALIZER pipelines including generating a detailed caption or scene graph as an intermediate step to PDDL formulation, compared with one VLMs-as-planner baseline. We evaluate each method over three vision-based datasets that drastically differ in difficulty and realism. On top of one existing visual Blocksworld dataset by Shirai et al. (2024b) that is small-scale, simulated, and fully observable via one single image, we propose a novel challenge dataset, BLOCKSWORLD-REAL, based on real images captured by sensors of physical robots. For each planning problem, we provide multiple photos from ego-centric viewpoints, challenging the model to identify and track objects across perspectives. These photos also closely resemble real-world visual conditions by including occlusion, motion blur, discoloration, and noisy background. We derive a similar multi-view planning dataset from ALFRED (Shridhar et al., 2020a), where the planning problem is described by multiple rendered images.

Our results position VLM-AS-FORMALIZER as a much stronger and more generalizable paradigm than end-to-end planning for long-horizon, visual-language planning. Even so, we attribute the still significant headroom to the VLMs' major weakness in visual detection rather than code generation, failing to capture an exhaustive set of objects and relations. While intermediate representations such as captions or scene graphs help, their inconsistent gain suggests future efforts.

## 2 PROBLEM FORMULATION

### 2.1 FORMAL DEFINITION

Formally, we define the *Vision-PDDL-Planning* task in line with established literature of STRIPS (Fikes & Nilsson, 1971) as follows (Figure 1). The input consists of a triplet $(V, I, \mathcal{D})$, where:

1. $V = v_1, \ldots, v_n$ is a sequence of $n$ images, with each image $v_i$ representing (possibly partial) observations of the initial environment;
2. $I$ is a natural language instruction specifying the goal;
3. $\mathcal{D}$ is a PDDL domain file, which formally defines the planning environment.

The domain file $\mathcal{D}$ provides the type system for entities, a set of relational predicates $R$, and a set of parameterized actions $A$. Each action is specified by its preconditions and effects, both expressed as conjunctive logical formulas over the predicates.

The final objective of the Vision-PDDL-Planning task is to generate a plan $L = [a_1(\bar{e}_1), \ldots, a_m(\bar{e}_m)]$, where each $a_i \in A$ is an action schema defined in the domain $\mathcal{D}$, and each $\bar{e}_i$ is a tuple of grounded entities corresponding to the parameters of $a_i$. The plan $L$, when executed in the environment represented by the images $V$, must achieve the goal specified by the instruction $I$.

While one could imagine an end-to-end model that maps the input triplet $(V, I, \mathcal{D})$ directly to a plan $L$, in this work we focus on a modular paradigm: VLM-AS-FORMALIZER. Specifically, this involves first generating a PDDL problem file $\mathcal{P}$, which encodes the initial state and goal extracted from $V$ and $I$, and then employing a PDDL solver to compute a valid plan.

At a high level, the problem file $\mathcal{P}$ is defined by three main components: objects ($E$), the initial state ($s_0$), and the goal state ($\psi$goal):

1. Objects: $E = e_1, \ldots, e_n$ is the set of named entities present in the environment. Each object is an instance of an entity type defined in $\mathcal{D}$.
2. Initial State: $s_0 \in \mathcal{S}$ is a set of relational facts of the form $r(\bar{e})$, where $r \in R$ is a relational predicate defined in $\mathcal{D}$; $\bar{e}$ is a tuple of entities from $E$ that participate in relation $r$.
3. Goal State: $\psi_{\texttt{goal}} : \mathcal{S} \to \mathbb{B}$ is a boolean formula over relational facts, where $S$ is the state space and $\mathbb{B} = \text{True}, \text{False}$. When $\psi_{\texttt{goal}}(s^*) = \text{True}$ we say that the state $s^*$ achieves the problem goal.

Here, the state space $\mathcal{S}$ comprises all possible configurations of relational facts over the object set $E$ with the predicates $R$. Each state $s \in \mathcal{S}$ is a set of instantiated facts $r(\bar{e})$, describing which relations

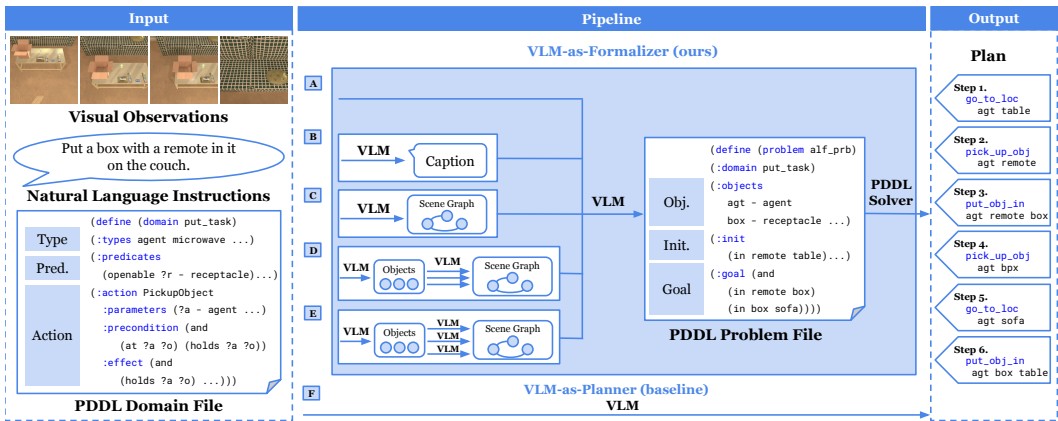

Figure 1: An illustration of the Vision-PDDL-planning task. The input includes visual observations of entities, natural language instructions of the goal, and a PDDL domain file. While a baseline might use a VLM to directly generate the plan, we advocate for the 5 VLM-AS-FORMALIZER pipelines that generates the problem file which deterministically derives the plan with a PDDL solver.

hold among the objects. A PDDL solver takes as input the domain-problem file pair $(\mathcal{D}, \mathcal{P})$ and search for a plan. If a plan $L$ is found, executing it from the initial state $s_0$ yields a sequence of intermediate states $s_1, s_2, \ldots, s^*$ such that the goal formula $\psi_{\text{goal}}(s^*)$ evaluates to True.

## 2.2 EVALUATION METRICS

To comprehensively measure the planning ability of VLM-AS-FORMALIZER methods, we propose two complementary sets of metrics: task-level metrics and scene-level metrics.

*Task-level* metrics evaluate the direct planning success of the method. Due to the nature of the Vision-PDDL-Planning task, there are three levels of success that a correct $\mathcal{P}$ needs to attain. The basic level of success is **compilation success** which is defined as a boolean function that evaluates to True if the PDDL solver can compile $\mathcal{P}$ into runnable code. The necessary and sufficient condition for compilation success is syntax correctness in $\mathcal{P}$. A level above is **planner success** which is defined as a boolean function that evaluates to True if the PDDL solver can find $L$ after exhaustive search. $\mathcal{P}$ needs to have non-contradicting initial states and goal states to achieve planner success. Finally, the most important metric is **simulation success**, which is defined as a boolean function that evaluates to True if the found $L$ can start from the initial states in the ground truth $\mathcal{P}$ to reach the target goal states. For each metric, we report the average success rate of all tasks in a benchmark as indicators for pipeline performance.

To provide finer-grained insights into the problem files generated by VLM-AS-FORMALIZER, we also consider *scene-level* metrics which evaluate the VLM's representation of the objects, initial states, and goals against a ground-truth $\mathcal{P}$, reporting **precision**, **recall** and **F1** for each category. A model with a low recall on any of the three categories will fail to find a plan, as the solver will not correctly satisfy all relevant conditions. A model with a high recall but a low precision might find a plan but will do so inefficiently, which is not accounted for by task-level metrics.

## 3 BENCHMARKS

In this section, we discuss the process to curate the benchmarks for VLM-AS-FORMALIZER.

### 3.1 CRITERIA

Before instantiating the formal definition and evaluation metrics above, we first consider the criteria of suitable benchmarks to study the effectiveness of VLM-AS-FORMALIZER in long-horizon, multimodal planning tasks. Minimally, they should include:

Figure 2: The three benchmarks in Vision-PDDL-Bench. Each benchmark provides visual observations (single or multiple images), a natural language instruction, and a PDDL domain file as input. The task is to generate a valid plan such that, after execution, the resulting state fulfills the goal expressed in the natural language instruction.

1. Visual observations $V$ and natural language goal instruction $I$ as input;
2. Ground-truth domain file $\mathcal{D}_{\text{domain}}$ as input, coupled with ground-truth problem file $\mathcal{P}_{\text{problem}}$ for evaluation.

To maximize real-life application, ideal benchmarks should be:

1. **Realistic** instead of simulated, as previous work (Hodaň et al., 2019; Wang et al., 2024b) has shown a systematic performance degradation of vision models working with photo-realistic instead of rendered images;
2. **Noisy** instead of clean, as previous work (Chen et al., 2018; Abdelhamed et al., 2020) has shown a systematic performance degradation of vision models working with poor lighting, motion blurring, discoloration, or occlusion between relevant objects.;
3. **Multi-view** instead of single-view, as recent work (Liu et al., 2024a; Wang et al., 2025) has faced the challenge of multi-image reasoning, though not in the context of planning;

The closest, existing benchmarks that fulfill the minimal criteria to our knowledge is the visual BLOCKSWORLD dataset from Shirai et al. (2024a). However, it falls short of the real-life criteria as it is **simulated** (rendered from a game engine), **clean** (under perfect lighting condition and visibility), and **single-view** (Figure 2). Moreover, it contains 10 examples with homogeneous initial conditions, potentially leading to overestimated and high-variance evaluation results.

## 3.2 PROPOSED BENCHMARKS

To address this issue, we propose two new benchmarks: BLOCKSWORLD-REAL based on a **realistic** staging of the classical Blocksworld domain (IPC, 1998) and ALFRED-MULTI based on the widely used ALFRED simulated environment (Shridhar et al., 2020a) (Figure 2). Naturally, both fulfill the minimal criteria with manually annotated and curated multimodal input and PDDL. They are both **noisy** under different extent of imperfect visual conditions by design, which more closely resemble the real-world environment where the robots are deployed. They are also both **multi-view**, providing multiple images that represent the same initial state of the environment from multiple perspectives. As a result, the same object can occur in multiple images, which poses a challenge for the VLM to identify them correctly and consistently. Often, the two benchmarks rather uniquely define a task environment that is *partially observable* with regard to each individual image, but *fully observable* only with regard to the complete set of images. Each image provides only a fraction of the necessary information for planning and cannot be relied on as the sole reference. Therefore, the VLM needs to extract information necessary for planning from all images.

We now describe the creation of the two benchmarks in more details. BLOCKSWORLD-REAL has the classic blocksworld setup: given an initial stacking of colored blocks, a model is required to create a new stacking of blocks as instructed. We collect a total of 102 problems using blocks of distinct colors. For each problem, which is generated programmatically, we also annotate the ground truth problem file $\mathcal{P}_{\text{gt}}$ that precisely describes the initial states and goal states of each block. We pair each problem file $\mathcal{P}_{\text{gt}}$ with an overarching domain file $\mathcal{D}_{\text{gt}}$ to a solver to get the ground truth plan. On average, each plan consists of 12 steps, which implies a large search space for long-horizon planning. To get the corresponding visual inputs of BLOCKSWORLD-REAL, we set up the exact initial stacking of each task using real blocks in a robotics lab and use a camera on a robotic arm to

Figure 3: Example pipeline of EP-SG-P. The prompting process involves first identifying the relevant objects, then predicting the existence of relations between them, and finally generating a PDDL problem file from the resulting scene graph. A PDDL solver is then used to produce the plan.

perform a sweep motion around the stacked blocks. This gives us one hundred images frames which we take four with equal temporal intervals to get a set of images with diverse viewpoints.

In contrast, ALFRED-MULTI simulates an indoor household environment where a model performs everyday tasks such as putting a clock on a desk or heating a meal in the microwave. We take 150 trajectories from the training split of the original ALFRED environment to compose our tasks. To obtain the ground-truth problem file $\mathcal{P}_{gt}$ and visual inputs, we reverse engineer the necessary object instances and object states that must occur in $\mathcal{P}_{gt}$ for the solver to find the exact same plan. As the necessary objects are already included in the ground-truth plan, to find each object's true grounded predicates, we take advantage of the provided PDDL file in each task to extract the relevant lines. Then, to get the minimally sufficient set of images taken from the environment, we feed the extracted object states into the provided simulation engine, and we collect only images that show at least one of the objects that occur in $\mathcal{P}_{gt}$. As a result, we now have a set of images which together define a fully observable environment of all relevant objects but individually define a partially observable environment of one or more objects. This yields on average four to six images per problem.

## 4 EXPERIMENTAL SETUP

### 4.1 METHODOLOGY

As is shown in Fig. 1, we consider 6 pipelines that use VLMs to produce a plan for the Vision-PDDL-planning task. Among them, the first five (A-E) fall into the category of VLM-AS-FORMALIZER, whereas the last (F) skips PDDL and generates the plan in an end-to-end manner. As is discussed in Section 2, all methods assume a common set of inputs, which consists of a list of visual observation images and a natural language goal instruction.

To begin with, the DIRECT-P pipeline (A) directly generates $\mathcal{P}_{pred}$ in a single call to the VLM. In the prompt, the VLM is first given an out-of-domain one-shot example of a problem file as a reference to the PDDL syntax. To ensure no semantic overlap with the current task, we use a toy example from the Tower of Hanoi domain. The VLM is also allowed to output intermediate reasoning steps though we do not explicitly prompt it to do so. We parse and extract $\mathcal{P}_{pred}$ from the output and input in into the solver to find the optimal plan.

To study the effect of scaling up test-time compute to explicitly analyze the scene, the CAPTION-P pipeline (B) first generates an intermediate **scene caption** in natural language and then the problem file $\mathcal{P}_{pred}$ in the second step. We prompt and require the VLM to generate scene captions that consist of five aspects of the scene: (i) relevant object types and their instances, (ii) the quantity of each object type, (iii) relevant spatial relationships between the objects, (iv) task-related object properties, and (v) vision-related object properties. The first two aspects force the VLM to accurately enumerate the objects that will serve as grounded arguments to the initial states in $\mathcal{P}_{pred}$. The third aspect directs the VLM to attend to the binary relations between objects in those states, whereas the fourth and fifth focuses on unary relations of each single object. Given the scene caption in the first step, the VLM is prompted again with the same input to output the $\mathcal{P}_{pred}$.

To enforce more formality on the model generation, the SG-P pipeline (C) instructs the VLM to generate a **scene graph** that describes the images. For each object type (e.g., `block`) and each predicate (e.g., `ontable(block)`) defined in the domain file $\mathcal{D}$, the model generates all the in-

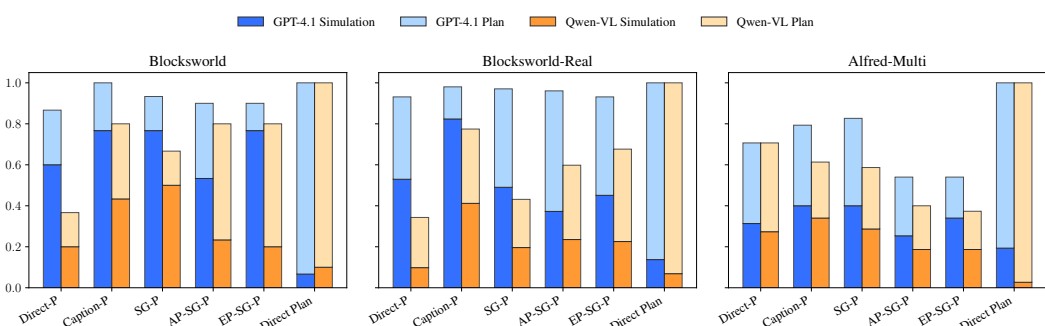

Figure 4: Success rates of six strategies across three benchmarks. Bars represent planner success (light), and simulator success (dark), reflecting increasing levels of strictness from syntax validity to plan generation to goal-achieving execution.

stantiated objects (e.g., red_blk:block) and grounded predicates (e.g., ontable(red_blk)). Given this scene graph, the second step is a pure translation task, where the model simply translates the instantiated objects and grounded predicates into $\mathcal{P}_{\text{pred}}$, in addition to predicting goal states based on the natural language instruction.

Concerned about possible low recalls of the grounded predicates in the scene graph generated by the previous two methods, we propose the AP-SG-P pipeline (D) that **eliminates false grounded predicates** from an exhaustive list of all possible grounded predicates. In the first pass, given the input, we first prompt the VLM to identify all relevant objects in the scene and their corresponding type. We then automatically enumerate all possible grounded predicates where each argument is an object instance identified by the VLM. In the second pass, we input **all** possible grounded predicates to the VLM at once to verify the existence each of the grounded predicates as 'True' or 'False' labels. The grounded predicates labeled as 'True' become the set of initial states that will go into $\mathcal{P}_{\text{pred}}$, and the identified object instances become the set of objects in $\mathcal{P}_{\text{pred}}$. In the final pass, we send the instruction along with all identified objects and initial states to the VLM to come up with the goal states which complete the $\mathcal{P}_{\text{pred}}$. The model is allowed to add or remove from the previously decided initial states based on a holistic understanding of the goal not previously accessible.

Similar to the above, the EP-SG-P pipeline (E) also first enumerates all possible grounded relations. However, to study the impact of context window size on the VLM's ability to identify the correct object relations, we iteratively pass **each** possible grounded relation, instead of all of them at once, to the VLM in a separate call. On our benchmarks, we find the increase in computation time is negligible when as the number of predicates is small.

Finally, as a baseline against all five VLM-AS-FORMALIZER approaches, we consider DIRECT-PLAN (F) to directly output a plan without an intermediate $\mathcal{P}_{\text{pred}}$. Along with a one-shot out-of-domain example of the structure of the plan as a sequence of grounded actions, we instruct the model to generate a plan that consists solely of grounded actions that are defined in $\mathcal{D}$.

## 4.2 INFRASTRUCTURE

We study two VLMs which are state-of-the-art in the open-source and proprietary domains, respectively. The proprietary model we study is GPT-4.1 (G-4.1), using the OpenAI client endpoint to call the 'gpt-4.1-2025-04-14' version of the model. The open-source model we study is Qwen2.5-VL-72B (Q-72B), which we host via vLLM on 4 NVIDIA H100 GPUs locally to enable fast inference. We use a temperature of 0.7 for both models and use a max token count of 1024 to avoid cutoff of $\mathcal{P}_{\text{pred}}$ generation. After the VLM predicts $\mathcal{P}_{\text{pred}}$, we pair it with the ground-truth $\mathcal{D}$ and employ the Fast Downward Planner (Helmert, 2006) to deterministically find the plan.

## 5 EMPIRICAL FINDINGS

We report an array of constructive findings resulting from the 6 pipelines on the 3 benchmarks.

| Model | Pipeline | Objects | | | Initial States | | | Goal States | | |
|---|---|---|---|---|---|---|---|---|---|---|
| | | P | R | F1 | P | R | F1 | P | R | F1 |
| **G-4.1** | DIRECT-P | 1.0000 | 0.6882 | 0.8153 | 0.7481 | 0.4604 | 0.5700 | 0.7500 | 0.6670 | 0.7060 |
| | CAPTION-P | 1.0000 | 0.7366 | 0.8483 | 0.7531 | **0.5085** | **0.6071** | 0.7612 | 0.7088 | 0.7341 |
| | SG-P | 1.0000 | **0.7582** | **0.8625** | 0.7283 | 0.4931 | 0.5880 | 0.7792 | **0.7163** | **0.7464** |
| | AP-SG-P | 1.0000 | 0.6018 | 0.7514 | 0.7920 | 0.4064 | 0.5372 | 0.7469 | 0.6018 | 0.6665 |
| | EP-SG-P | 1.0000 | 0.6056 | 0.7544 | 0.7316 | 0.4191 | 0.5329 | 0.7308 | 0.5930 | 0.6547 |
| **Q-72B** | DIRECT-P | 1.0000 | 0.4367 | 0.6080 | 0.6703 | 0.2126 | 0.3228 | **0.9171** | 0.4246 | 0.5805 |
| | CAPTION-P | 1.0000 | 0.5661 | 0.7230 | 0.7459 | 0.3716 | 0.4960 | 0.8446 | 0.5350 | 0.6551 |
| | SG-P | 1.0000 | 0.4370 | 0.6082 | 0.7237 | 0.2492 | 0.3708 | 0.9017 | 0.4149 | 0.5683 |
| | AP-SG-P | 0.4485 | 0.2676 | 0.3352 | 0.3754 | 0.2138 | 0.2724 | 0.3991 | 0.2610 | 0.3156 |
| | EP-SG-P | 0.6250 | 0.2145 | 0.3193 | 0.4191 | 0.1323 | 0.2012 | 0.5576 | 0.2050 | 0.2997 |

Table 1: We assess the quality of generated PDDL problem files along three dimensions: correctly identified objects, correctly predicted initial states, and correctly specified goal states. We evaluate these aspects using Precision (**P**), Recall (**R**), and **F1** across all VLM-AS-FORMALIZER pipelines.

**VLM-AS-FORMALIZER establishes a strong advantage over end-to-end planning.** Across all three benchmarks and two VLMs (Figure 4), DIRECT-PLAN achieves close-to-zero performance, suggesting that even state-of-the-art VLMs are unable to reliably generate plans in multimodal, long-horizon planning tasks. In contrast, the five VLM-AS-FORMALIZER pipelines consistently achieve superior performance, with the least performing pipeline EP-SG-P still gaining a considerable advantage. The clear superiority of VLM-AS-FORMALIZER stands in contrast with previous work in text-based planning, where end-to-end LLM-as-planner pipelines often lead to effective results on complex domains (Huang & Zhang, 2025). This can likely be explained by VLMs' lack of inference-time scaling of reasoning tokens as some LLMs do, which are crucial for solving such high-complexity tasks. Comparing the performance on our proposed BLOCKSWORLD-REAL and the existing simulated BLOCKSWORLD benchmark, the added realism, multiple views, and degraded image quality add challenges to the VLM-AS-FORMALIZER methods, while the performance of certain pipelines remains robust.

**VLM-AS-FORMALIZER benefits from generating intermediate representations.** We observe CAPTION-P generating captions and SG-P generating scene graphs consistently outperform DIRECT-P generating PDDL directly, across benchmarks on both planner and simulation success rates. The improvements are most pronounced on BLOCKSWORLD and BLOCKSWORLD-REAL. The competitive advantage carries over to better precision and recall in $\mathcal{P}_{\text{pred}}$, as shown in Table 1. Together, this suggests that both methods "see better" or attain more successes on visual grounding by generating an intermediate representation, despite harnessing the same perceptual capacity of the model, showing the promise of inference scaling on vision. No advantage is observed from more complex inference techniques proposed in AP-SG-P and EP-SG-P .

**The bottleneck of VLM-AS-FORMALIZER is visually grounding initial states.** Recall that the objective of a VLM-AS-FORMALIZER pipeline on the Vision-PDDL-Planning task is to generate a problem file. While not shown in Figure 4, the compilation success rate for all pipelines on all datasets is 100%, suggesting feasibility of generating syntactically correct PDDL. Semantically, the problem file consists of three components: objects, initial states, and goal states. In our task formulation, the objects and initial states are solely informed by the visual input, while the goal states are solely inform by the textual input. As reported in Table 1, the F1 scores of initial state predictions in $\mathcal{P}_{\text{pred}}$ are significantly lower than those of object and goal state predictions across models and pipelines. The discrepancy suggests that VLMs' incapability of object relation detection, rather than language understanding, is the primary bottleneck.

**VLMs are more prone to omit correct states than proposing incorrect ones.** All pipeline's predictions of objects, initial conditions, and goals show consistently lower recall than precision, highlighting VLMs' struggles with false negatives. The right section of Figure 6, showing SG-P results, illustrates a common cause of false negatives. In the example, to successfully predict all relevant states of a block being on the table and with no other blocks on its top, the VLM needs to

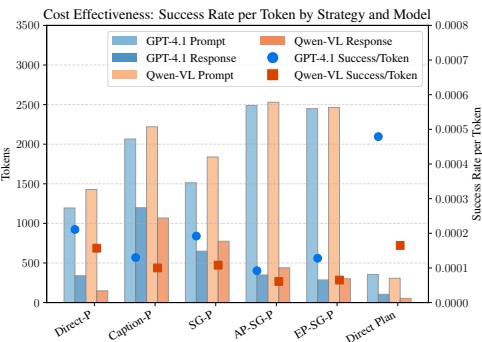

Figure 5: Cost effectiveness of LLM-based planning strategies. The stacked bars report the average number of prompt and response tokens consumed by GPT-4.1 and Qwen-VL under each strategy. Overlaid markers indicate the simulation success rate normalized by the amount of tokens.

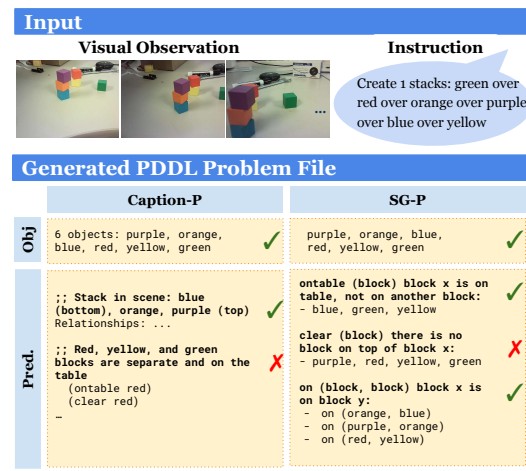

Figure 6: Examples of generated captions and scene graphs on BLOCKSWORLD-REAL.

predict the block x as being both in the state (clear x) and in (ontable x). However, it is a common pitfall for VLMs to predict only one of the two, leading to planner failure when the PDDL solver cannot locate x to enable the search.

**Intermediate representations affect how VLMs perceive the scene.** As shown in Figure 6, which compares $\mathcal{P}_{\text{pred}}$ generated by CAPTION-P and SG-P , the different intermediate representations induced by prompt strategies in the two pipelines lead to significant drift in the content of $\mathcal{P}_{\text{pred}}$. The prompt of CAPTION-P is *structure-oriented*, guiding the VLM to first perceive objects and then capture their organization into patterns such as stacks of blocks, rather than describing objects in isolation. This perceptual pattern is reflected in the resulting $\mathcal{P}_{\text{pred}}$ which accurately grounds a stack of blocks (blue, orange, purple) but completely mischaracterizes another stack (red, yellow). By contrast, the prompt of SG-P is *relation-centric* in the sense that it asks the VLM to iteratively ground each predicate in $\mathcal{D}_{\text{gt}}$, which takes multiple distinct objects as arguments. This results in the VLM successfully grounding a complete set of relations sharing the same predicate (ontable) but grossly misses others (clear, on). At inference time, the prompting strategies of different pipelines lead to non-trivial disagreements in the VLM's perception of relational facts, causing real differences in simulation and planning success.

**The planning superiority of VLM-AS-FORMALIZER is a tradeoff with token efficiency.** We plot the number of tokens generated by each pipeline averaged across benchmarks in Figure 5, coupled with the average success rate per token, calculated by dividing the average success rate by the number of total tokens in a single task. We witness that DIRECT-PLAN significantly outperforms all VLM-AS-FORMALIZER methods in terms of token efficiency, the reverse of what we observed when judging success rates. Among VLM-AS-FORMALIZER pipelines, CAPTION-P, consistently the leading method in terms of planning success, ranks among the least token-efficient approaches. Although it enjoys generally 10% to 30% more planning successes than DIRECT-P, it also consumes more than 102% of the total tokens through verbalization of a set of perceptual tasks. The pronounced differences between pipelines point to a frontier for exploring improvements in VLMs' inherent formalizer capacity, with DIRECT-P striking a balance under current VLM capabilities.

## 6 RELATED WORK

**LLM as Planners and Formalizers** In purely textual planning, LLMs have been extensively studied both as planners (Wei et al., 2025) and as formalizers (Tantakoun et al., 2025). As planners, LLMs have been reported to perform strongly on short-horizon planning (Huang et al., 2022; Hu et al., 2023; Ahn et al., 2022) but their performance degrades on complex, long-horizon planning

tasks (Valmeekam et al., 2023b; 2025). As formalizers, LLMs have been posed to deliver increased robustness and interpretability (Lyu et al., 2023; Zhao et al., 2023; Guan et al., 2023) though such benefit has been partially verified and partially questioned on planning tasks with increased complexity (Zuo et al., 2025; Huang & Zhang, 2025; Kagitha et al., 2025). Regardless, studies of either method on multimodal data has been limited.

**VLM as Planners and Formalizers**   Some preliminary steps have been taken on formalizer approaches in the vision domain (Radford et al., 2021; Huang et al., 2023a; Ahn et al., 2022), yet current success is often shown on multimodal tasks with the aid of non-vision inputs (Kwon et al., 2025; Li et al., 2025; Liang et al., 2025). For tasks that focus only on vision, they rely mainly on close-vocabulary (Shirai et al., 2024b; Siburian et al., 2025), few-shot techniques (Herzog et al., 2025b) in overly simplified scenarios (Dang et al., 2025). More broadly, techniques that reason over text-and-image inputs attempt to grounded object information by identifying visible objects in the current scenario (Song et al., 2023; Huang et al., 2023b), and employ caption-based approaches (Yang et al., 2025a). More advanced methods utilize structured symbolic representations, such as scene graphs in both 2D and 3D scenarios, to improve reasoning capabilities (Herzog et al., 2025a; Mitra et al., 2024; Wang & Liu, 2024; Jiao et al.; Zhu et al., 2021), and develop open-domain scene graph grounding models (Zhang et al., 2025; Gu et al., 2024).

**Multimodal Planning Benchmarks**   A series of works has been established for evaluating embodied agents' performance, spanning tasks from high-level representation understanding (Shridhar et al., 2020a; Li et al., 2023; Shirai et al., 2024a), to fine-grained, low-level continuous control (Zheng et al., 2022; Khanna et al., 2024; Zhang et al., 2024). These evaluations cover diverse application domains such as object manipulation (Zheng et al., 2022; Ahn et al., 2022; Valmeekam et al., 2023a), household tasks (Shridhar et al., 2020b; Liu et al., 2024c; Merler et al., 2025), and navigation scenarios (Gadre et al., 2023; Jain et al., 2024), progressively evolving from artificial, structured tasks toward more realistic challenges relevant to human activities. Moreover, there is an increasing shift from purely simulated environments (Kolve et al., 2017; Szot et al., 2021) toward real-world robotic deployments (Zitkovich et al., 2023; Driess et al., 2023; Khazatsky et al., 2024). Evaluation settings have shifted also from primarily single-view, fully observable symbolic scenarios (Shridhar et al., 2020a; Valmeekam et al., 2023a; Li et al., 2023) to more complex, multi-view setups with partial or complete observability (Yan et al., 2018; Khazatsky et al., 2024), reflecting greater realism in embodied agent interactions.

**Open-Vocabulary Detection and Scene Graph Generation**   The ability to detect and localize objects described in natural language has become increasingly important for vision-language tasks. With an evolving level of sophistication over earlier works (Kazemzadeh et al., 2014; Yu et al., 2018; Wu et al., 2022), the emergence of large-scale vision-language pretraining, such as CLIP (Radford et al., 2021), GLIP (Li et al., 2022), and Grounding DINO (Liu et al., 2024b), have achieved remarkable performance. In addition to object detection, scene graphs provide structured representations of visual scenes. Pioneering work by Lu *et al.* (Lu et al., 2016) and Xu *et al.* (Xu et al., 2017) introduced visual relationship detection through language priors, whose key challenges have been subsequently addressed (Zellers et al., 2018; Tang et al., 2020; Knyazev et al., 2020). Recent advances have explored knowledge-embedded approaches (Chen et al., 2019), natural language supervision (Zhong et al., 2021), and temporal dynamics in video scenes (Cong et al., 2021). While traditional scene graph generation focuses on single-image parsing, our work requires VLLMs to construct scene graphs from multiple partial views and translate them into formal logical representations for planning. This poses unique challenges in cross-view consistency and the integration of spatial reasoning with symbolic planning languages.

## 7   CONCLUSION

In this work, we evaluated the effectiveness of VLM-AS-FORMALIZER in solving long-horizon visual-language planning tasks. We proposed two novel benchmarks, BLOCKSWORLD-REAL and ALFRED-MULTI, that for the first time present partially observable, multi-view environments in the Vision-PDDL-Planning domain. Our evaluation of five VLM-AS-FORMALIZER pipelines demonstrates their effectiveness over the end-to-end baseline and establishes VLMs as blind thinkers whose

success depends on a crucial interplay of vision and language. Limitations in current approaches, such as token inefficiency and insufficient recall, remain to be addressed by upcoming research.

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

# A  PROMPT TEMPLATES

We provide the complete set of prompt templates for each of the six pipelines defined in section 4.

## A.1  PROMPT FOR DIRECT-P

```
You are helping a robotic planning task.
Given the image of a scene and an instruction, generate the PDDL
    file with objects, initial state, and goal specification.
Your output should adhere to the constraints defined in the domain
     file.
You must output the PDDL file in the correct format.
Examples of PDDL problems given a different domain instruction:
{{PDDL_problem_file_for_hanoi}}

For the current domain, this is the domain file:
{{PDDL_domain_file}}

The image of the scene has been provided.
Please first analyze the image and then generate the PDDL problem.

{{environment_specifics}}
Instruction: {{instruction}}
```

## A.2  PROMPT FOR DIRECT-PLAN

```
You are helping a robotic planning task.
Given the image of a scene and an instruction, generate a step-by-
    step plan for the robot(s).
All the possible actions and their arguments are given below:
{{actions_with_signatures}}

You must first reason what objects are in the scene that can be
    the arguments of the actions.
Then you must reason what actions to take in the plan. Be mindful
    of the preconditions and effects of the actions.
Each action should take one line in the plan.
The plan should be in the following format:
(action1 arg1 arg2 arg3)
(action2 arg1 arg2 arg3)
...

For the current domain,
{{environment_specifics}}

The image of the scene has been provided.
Please first analyze the image and then generate the plan.
Do not generate anything after the plan.

Instruction: {target["instruction"]}
Please generate the plan for the robot. Do not generate anything
    after the plan.
```

## A.3 PROMPTS FOR CAPTION-P

**Step 1: Caption generation**

```
You are helping a robotic planning task.
Given the image of a scene and an instruction, generate a detailed
    scene description.

For the current domain, this is the domain file:
{{PDDL_domain_file}}

{{environment_specifics}}
Instruction: {{instruction}}

Analyze the provided images and generate a detailed scene
    description that focuses on:
1. Objects and their types (especially those defined in the domain
    above)
2. Quantities of each object type
3. Spatial relationships between objects (particularly those
    matching domain predicates)
4. Current state of each object relevant to the planning task
5. Colors, positions, and other visual properties that help
    distinguish objects

IMPORTANT:
- Focus on objects that are relevant to the domain types and the
    given instruction
- Describe relationships that match the predicates defined in the
    domain
- Be specific about object names and their distinguishing features
- Only generate a scene description, do NOT generate any PDDL

Provide a comprehensive scene description:
```

**Step 2: Problem file generation**

```
You are helping a robotic planning task.
Given the image of a scene and an instruction, generate the PDDL
    file with objects, initial state, and goal specification.
Your output should adhere to the constraints defined in the domain
    file.
You must output the PDDL file in the correct format.
Examples of PDDL problems given a different domain instruction:
{{PDDL_problem_file_for_hanoi}}

For the current domain, this is the domain file:
{{PDDL_domain_file}}

    The image of the scene has been provided.
    Please first analyze the image and then generate the PDDL
        problem.

{{environment_specifics}}
Instruction: {{instruction}}

SCENE CAPTION:
{{caption}}
```

```
Based on the above scene caption, the images, and the instruction,
    generate the PDDL problem file.
Use the caption to understand what objects are present and their
    current state.
Make sure the PDDL problem accurately reflects the scene described
    in the caption.

Generate the PDDL problem:
```

## A.4 PROMPTS FOR SG-P

**Step 1: Scene graph generation**

```
You are helping a robotic planning task.
Given the image of a scene and an instruction, generate the PDDL
    file with objects, initial state, and goal specification.
Your output should adhere to the constraints defined in the domain
    file.
You must output the PDDL file in the correct format.

For the current domain, this is the domain file:
{{PDDL_domain_file}}

The image of the scene has been provided.
You must first generate a scene graph for the image, using the
    types and predicates in the domain file.
Then use the scene graph to generate the PDDL problem.
Do not generate anything after the PDDL problem.

Important: When listing objects, provide descriptive names that
    include relevant visual characteristics that would help
    identify them in the image. Use specific, descriptive object
    names rather than abstract or generic terms.

Format:
Scene graph:
{obj_type}: describe all objects of this type in the image.
{predicate} ({arg_type}) {comment}: describe all tuples (predicate
    , object) that satisfy the predicate.
{predicate} ({arg1_type}, {arg2_type}) {comment}: describe all
    tuples (predicate, object1, object2...) that satisfy the
    predicate.

PDDL problem: <PDDL problem>

{{environment_specifics}}
Instruction: {{instruction}}

IMPORTANT:
1. Generate ONLY the scene graph based on the images
2. Do NOT generate any PDDL problem file
3. Stop immediately after completing the scene graph
4. Follow the format shown above for scene graph generation
5. You must ONLY use the predicates defined in the domain file
    above. Do not invent new predicates or use predicates not
    explicitly listed in the domain.
```

**Step 2: Problem file generation**

```
You are helping a robotic planning task.
Given the image of a scene and an instruction, generate the PDDL
    file with objects, initial state, and goal specification.
Your output should adhere to the constraints defined in the domain
     file.
You must output the PDDL file in the correct format.
Examples of PDDL problems given a different domain instruction:
{{PDDL_problem_file_for_hanoi}}

For the current domain, this is the domain file:
{{PDDL_domain_file}}

    The image of the scene has been provided.
    Please first analyze the image and then generate the PDDL
        problem.

{{environment_specifics}}
Instruction: {{instruction}}

The following scene graph has been generated from the image
    analysis:

Scene Graph:
{{scene_graph}}

Based on this structured scene graph and the images provided,
    generate the PDDL problem file.

Requirements:
1. All objects mentioned in the PDDL must exist in the scene graph
2. All predicates used must be consistent with the scene graph
    analysis
3. The initial state must reflect the current scene as described
    in the scene graph
4. The goal state must align with the given instruction
5. Generate a complete and properly formatted PDDL problem file
6. Avoid repetitive or contradictory statements

Generate the PDDL problem:

IMPORTANT: You must ONLY use the predicates defined in the domain
    file. Do not invent new predicates.
```

## A.5 PROMPTS FOR AP-SG-P AND EP-SG-P

**Step 1: Object detection**

```
You are given some images which contain various objects of
    interests for a given task.

The following domain file specifies all possible states and
    actions for the task:
{{PDDL_domain_file}}

Given the name of an object type, identify all objects with an
    appropriate name in the images that belong to this type.
Follow this exact format:
<object> - <type>
<object> - <type>
...

The images have been provided. {{environment_specifics}}
The task instruction is: {{instruction}}
Now identify all the objects for the following types relevant to
    the task instruction:
{{object_type_1}}:
{{object_type+2}}:
```

**Step 2 (AP-SG-P): Batched relation detection**

```
Answer 'yes' or 'no' for each of the following statements:

1. Is {{relation_description}}
2. Is {{relation_description}}
3. Is {{relation_description}}
...
{n}. Is {{relation_description}}

Provide your answers in order, one per line, using only 'yes' or '
    no'.
```

**Step 2 (EP-SG-P): Individual relation detection**

```
Is {{relation_description}}
Answer with only 'yes' or 'no'.
```

**Step 3: Goal formalization**

```
You are given some images which contain various objects of
    interests for a given task.
The images have been provided. {{environment_specifics}}
The task instruction is: {{instruction}}

The following domain file specifies all possible states and
    actions for the task:
{{PDDL_domain_file}}

The following are all the objects and their states:
{{detected_objects}}
The following are all the initial states:
{{detected_relations}}
```

```
For the task instruction, {{instruction}}, generate the goal
    specification for the PDDL file.
The goal specification should be in the following format:
(:goal (and
    (predicate arg1 arg2)
    (predicate arg)
    ...
  )
)
```

