# OpenReview forum: "Are Vision Language Models Blind Thinkers on Zero-Shot Multimodal Planning Tasks?"
_ICLR.cc/2026/Conference — Submitted to ICLR 2026_

### Official Review · Reviewer_m2AE · 2025-10-31

**Soundness:** 3
**Presentation:** 4
**Contribution:** 3
**Rating:** 6
**Confidence:** 2

**Summary:**

The authors explore the application of the LLM-as-formalizer paradigm to VLMs. They consider 2 VLMs (GPT4.1 and Qwen-VL) and 5 VLM-as-formalizer pipelines. They introduce two novel multimodal benchmarks by annotating ground truth PDDL files for Blocksworld-Real and Alred-Multi. The evaluation considers 3 levels of success: 1st) executable PDDL syntax, 2) solver can find a plan with exhaustive search and 3rd) executing the plan leads to a success in the simulation. They find that VLM-as-formalizer is particularly helpful in long horizon tasks but visual capabilities remain the main bottleneck.

**Strengths:**

- well designed benchmark setup with well motivated design decisions
- useful adaption of existing datasets for VLM-as-formalizer
- Exhaustive related work section

**Weaknesses:**

- novelty is limited but great starting point for future research in this direction

**Questions:**

- Does the max token count of 1024 mentioned in 4.2. include thinking tokens of the VLM?

---

> ### Author Response · Authors · 2025-11-18
> **Novelty of the VLM-As-Formalizer Framework**
>
> While the implementation of each **pipeline** is straightforward, we would like to point out that the VLM-as-formalizer **methodology** is a novel and promising concept as we have clearly argued in Section 1. In past ICLR oral presentations, we consistently observe works that similarly proposed novel methodologies with thorough empirical evaluations despite simple pipelines, such as [1, 2, 3]. We argue that our comprehensive evaluation of practical pipelines is itself a contribution merited of an acceptance.
>
> [1] Yue et al., Inference Scaling for Long-Context Retrieval Augmented Generation, 2024 (ICLR 2025 Oral)
>
> [2] Weng et al., Do as We Do, Not as You Think: the Conformity of Large Language Models, 2025 (ICLR 2025 Oral)
>
> [3] Song et al., Mind the Gap: Examining the Self-Improvement Capabilities of Large Language Models, 2025 (ICLR 2025 Oral)

---

### Official Review · Reviewer_BYka · 2025-10-31

**Soundness:** 3
**Presentation:** 3
**Contribution:** 3
**Rating:** 6
**Confidence:** 4

**Summary:**

This paper introduces 5 VLM-as-Formalizer (\vaf) pipelines for one-shot, open-vocabulary, multimodal PDDL formalization, evaluated on two existing and two new datasets (Blocksworld and ALFRED) with realistic visual conditions. Results show \vaf surpasses end-to-end plan generation and highlights that current VLMs are constrained more by vision than language, particularly in modeling complete object relations. Scene graph–based inference offers partial mitigation, pointing to future directions in visual PDDL generation.

**Strengths:**

1. New Benchmarks: The paper’s key strength lies in introducing the Blocksworld and ALFRED datasets, which move beyond oversimplified benchmarks. By incorporating real photos, multi-view scenes, occlusion, and motion blur, these datasets offer a much more realistic and impactful testbed for multimodal planning.

2. The results strongly support the idea that current VLMs reason well but see poorly. Low F1 scores in visual grounding versus text-based grounding (Table 1) clearly reveal vision, not language, as the main bottleneck—an insightful and memorable finding.

3. Instead of a single model, the paper systematically compares five formalizer pipelines, showing that intermediate representations (e.g., captions, scene graphs) significantly outperform direct PDDL generation. This highlights that inference-time reasoning can partly offset visual limitations.

**Weaknesses:**

1. Token Inefficiency: Figure 5 shows that top-performing pipelines like CAPTION-P achieve higher accuracy but at a huge token cost, making them far less efficient than direct planners. This limits scalability and practicality for real-world, resource-constrained systems.

2. Limited Scope of “Thinking”: The “blind thinker” metaphor somewhat exaggerates the VLM’s role—its function is mainly to translate scenes into PDDL state descriptions, while actual multi-step planning is handled by an external solver. The model formalizes states, not plans.

**Questions:**

All of my qeustions are listed in the weakness section. If my concerns are well addressed, I will raise my rating.

---

> ### Author Response · Authors · 2025-11-18
> **Practicality Concerns of Pipelines**
>
> While the reviewer correctly points out that Caption-P is not optimal for token efficiency, we want to reaffirm that the practicality of this work lies not in presenting a globally optimal pipeline but in thoroughly analyzing different pipelines to guide future research direction.
>
> The discovered fact that Caption-P is not token efficient is one such contribution, as it calls for potential directions to either budgeting the reasoning length of Caption-P or scaling up training for efficient approaches such as Direct-P.

---

> ### Author Response · Authors · 2025-11-18
> **Updates on Title Scope**
>
> We appreciate the reviewer's diligence in the paper title, and agree that the VLM is playing a formalizing not reasoning role. We have thus updated the title to "Vision Language Models Cannot Plan, but Can They Formalize?" in the new upload.

---

### Official Review · Reviewer_q21t · 2025-11-01

**Soundness:** 2
**Presentation:** 2
**Contribution:** 2
**Rating:** 4
**Confidence:** 3

**Summary:**

This paper systematically investigates the paradigm of using Vision Language Models (VLMs) as PDDL formalizers (VLM-AS-FORMALIZER) for long-horizon, multimodal planning tasks, in contrast to end-to-end plan generation. The authors devise five modular pipelines for open-vocabulary PDDL problem generation, benchmarked against a VLM-as-planner baseline. Evaluation covers existing datasets and two new challenging, realistic, multi-view benchmarks (BlockSWORLD-REAL and ALFRED-MULTI) emphasizing imperfect real-world visual conditions. Extensive empirical results, supported by quantitative and qualitative analysis, reveal strengths and persistent limitations in VLM-based multimodal planning, especially in visual groundings.

**Strengths:**

- **Systematic, Comparative Framework:** The paper provides a thorough and well-structured empirical comparison of five distinct VLM-AS-FORMALIZER pipelines and a planner baseline, all on the identical, ambitious long-horizon planning task (Section 4.1 and Figure 1). This comprehensiveness goes beyond much prior work, which often examines a single method in isolation or under-simplifies visual scenarios.
- **Realistic Benchmarking:** The introduction of BlockSWORLD-REAL and ALFRED-MULTI as multi-view, real-image, challenging planning benchmarks (Section 3.2, Figure 2) advances evaluation realism and addresses key shortcomings in prior datasets (single-view, synthetic, or trivially structured). This enables new insights regarding VLM capabilities in noisy, partially observed environments.
- **Detailed Analysis and Insights:** Both quantitative (Table 1, Figure 4, Figure 5) and qualitative (Figure 6) evaluations are presented, and the discussion draws sharp, data-driven conclusions about why and where VLMs fail—especially the gap in visual scene grounding versus goal and object recall, and the role of prompt/intermediate representation strategy.
- **Mathematical and Technical Rigor:** The paper offers a precise formal definition of the Vision-PDDL-Planning task (Section 2.1), specifying entity sets, state representations, and evaluation criteria. This provides a sound basis for reproducibility and deeper analysis.
- **Clarity in Limitations:** The authors do not shy from articulating where VLMs fall short (e.g., "blind thinkers" bottlenecked by vision). Failure patterns are meticulously mapped in the results (Table 1, Figure 6), helping inform where future research should target.
- **Comparative Token Efficiency Analysis:** By introducing token efficiency metrics (Figure 5), the paper highlights important tradeoffs relevant to practical deployment in resource-constrained settings.

**Weaknesses:**

1. **Limited Novelty in Core Paradigm:** The VLM-as-formalizer concept—and generating PDDL from multimodal input—has close precedent in several recent works, including Shirai et al. (2024), Kwon et al. (2025), Dang et al. (2024), and multiple others reviewed in the related work section. The degree of novelty primarily lies in the systematic empirical benchmarking and pipeline analysis, not in radically advancing the methodology itself.
2. **Weaknesses in Related Work Positioning:** Several highly relevant directly related works (see “Potentially Missing Related Work” below) are not cited or discussed, notably those that handle similar formalization tasks or propose comparable systems (e.g., Image2PDDL or reflective planning with VLMs). For instance, Dang et al. (2024)'s 'Image2PDDL' mirrors aspects of the problem definition but is only implicitly addressed. The broader context of reflective or structured VLM planning is underexplored.
3. **Insufficient Baseline/Method Diversity:** The baseline and compared pipelines all involve either single VLMs or simple prompting variations. Despite claims of zero-shot and open-vocabulary capabilities, there is little inclusion or discussion of advanced compositional, neuro-symbolic, or hybrid methods that could integrate external object detectors, or explicit multi-step visual reasoning, as done in some related works or through more advanced architectural solutions (e.g., see Gu et al., 2024; Jiao et al., 2022). More diversity in competing approaches is desirable.
4. **Shallow Theoretical Analysis:** Beyond basic formalization, the paper provides little insight or formal modeling of the failure modes. For example, there is no mathematical dissection of how visual ambiguity or multi-view inconsistency propagates through PDDL generation. While the F1 breakdown in Table 1 is informative, deeper theoretical or optimization analysis—such as loss decomposition or an explicit error budget—would improve the scientific rigor.
5. **Equation and Notation Clarity:** The paper gives a succinct formal statement of the task (Section 2.1), but does not specify sampling/selection strategies for negative versus positive object relations, handling of ambiguous vision cues, or precise mechanisms for entity disambiguation across views in its mathematical form. For instance, it is unclear how VLMs resolve conflicts or occlusions in entity matching—critical for applications relying on cross-image consistency.
6. **Qualitative Analysis Covers Only a Few Representative Cases:** While Figure 6 provides qualitative illustration, these are limited to BlockSWORLD-REAL and capture only a small subset of error types. Broader visualization across datasets and tasks—and perhaps a taxonomy of common errors—would make the findings more robust.
7. **Reproducibility and Implementation Details:** Although the high-level approach and evaluation metrics are well described, many practical details are not spelled out, such as the prompt wording, exact hyperparameters for each VLM pipeline, and the PDDL parsing/compilation protocol. Additionally, there is no statement regarding code/data release, which impedes open comparison.
8. **Reliance on Proprietary Models and Limited External Validation:** GPT-4.1 performance may not generalize to non-commercial settings, and only one open-source model is tested. There is no systematic sensitivity analysis for model version, size, or training data variations. The findings could benefit from validation on a broader suite of models or architectures.
9. **Token Efficiency Framing Could Be Deepened:** While Figure 5 is a welcome inclusion, the discussion does not explore whether token count differences actually impact planning failure/intermediate errors, or if efficient summarization tricks would yield comparable performance.
10. **No End-to-End Embodied Agent Evaluation:** The pipelines are validated only through planning success, not actual execution on embodied agents in closed-loop settings. This reduces the operational significance of the findings, as simulation performance may not fully mirror real-world embodied planning challenges.
11. **Qualitative Differences Between Pipelines Not Probed Deeply Enough:** For instance, the differences in intermediate representations between CAPTION-P and SG-P are discussed, but the underlying cognitive/attention or error propagation mechanisms are not explored in depth, nor are possible remedies tried.

## Potentially Missing Related Work

1. **Yang, Z., Garrett, C. R., Fox, D. (2024): "Guiding Long-Horizon Task and Motion Planning with Vision Language Models"**
   - Directly integrates VLMs with hierarchical task and motion planning for long-horizon tasks, mirroring the vision-language-planning intersections here. Should be discussed in Section 6 and compared as a baseline or reference pipeline in empirical sections.
2. **Shirai, K., Beltran-Hernandez, C. C., Hamaya, M. (2024): "Vision-Language Interpreter for Robot Task Planning"**
   - Proposes using VLMs for generating problem descriptions for robotic task planning—directly paralleling VLM-as-formalizer logic. Needs explicit discussion and comparison in the related work section and empirical pipeline design.
3. **Feng, Y., Han, J., Yang, Z. (2025): "Reflective Planning: Vision-Language Models for Multi-Stage Long-Horizon Robotic Manipulation"**
   - Introduces test-time computation enhancements for VLM-based planning, relevant to the inference scaling ideas here. Should be referenced and positioned relative to novel aspects of the pipelines introduced.
4. **Dang, X., Kudláčková, L., Edelkamp, S. (2024): "Planning with Vision-Language Models and a Use Case in Robot-Assisted Teaching"**
   - Presents the 'Image2PDDL' framework, which closely aligns with the paper's formalization goals. Should be cited and compared in both the methodology and empirical results.

**Questions:**

1. **Entity Disambiguation and Multi-View Consistency:** How exactly does each pipeline resolve ambiguities or conflicts when the same entity appears in multiple images (e.g., under occlusion or motion blur)? Are there explicit mechanisms for cross-image entity fusion, or is this delegated entirely to VLM's latent capacity? Further details here—both in text and formula—would clarify the implementation.
2. **Negative Sampling and Initial State Recall:** Section 5 and Table 1 highlight that initial state recall is a primary failure point. Could the authors share which negative or ambiguous object relations are most frequently missed, perhaps with additional figures or even a confusion matrix? Would a different loss formulation or pipeline targeting high-recall make a significant difference?
3. **Prompt Engineering and Pipeline Robustness:** Are the observed differences in performance between CAPTION-P, SG-P, etc., mainly due to prompt wording or the inherent structure of intermediate representations? Is there evidence that better prompts would allow the simpler pipelines (e.g., DIRECT-P) to catch up? Sharing prompts or performing ablation over prompt structure would be valuable.
4. **Choice of Models:** Why was only GPT-4.1 and Qwen2.5-VL72B chosen? Were other VLMs or multimodal models considered, and if so, how did results compare, especially in lower-resource or real-time settings?
5. **Code/Data Availability:** Will the benchmarks (BlockSWORLD-REAL and ALFRED-MULTI), pipeline code, and evaluation scripts be publicly released to promote reproducibility and follow-up research?
6. **Full Embodied Evaluation:** Do the findings on planning success translate to real-world embodied agent execution, or is there a substantial sim2real gap? Any evidence (even anecdotal) here would be useful.

---

> ### Author Response · Authors · 2025-11-18
> **Considerations of New Related Works**
>
> We appreciate the reviewer's diligence in finding literature related to our work. We would like to point that, as for Shirai et al. 2024, we have not only included it in related works in section 6, but have also thoroughly evaluated on the blocksworld dataset it proposes as part of the core benchmark in section 3.
>
> As for Dang et al. 2024, we agree that it resembles the pipeline Direct-P, despite having simple, 2-dimensional, synthetic settings for all tasks and lacking comparisons between pipelines. We have added the citation to this work to our Section 6.
>
> For the two remaining works, we find the other cited works to be tangential to the topic of this paper and thus withhold from adding them to related works. Yang et al. 2024 is targeting sub-goal generations in close-vocabulary, synthetic settings, and therefore does not align closely with the focus on open-vocabulary, PDDL generation in this work. Similarly, Feng et al. 2025 is discussing an inference-time technique for VLM-as-planner frameworks, therefore bearing little relevance to our investigations on VLM-as-formalizer methods.

---

> ### Author Response · Authors · 2025-11-21
> **Clearly AI-generated reviews; will not respond to the rest**
>
> Apart from the discussions on related work, which we believe we have addressed above, we will not respond to the rest of the weaknesses and questions which are clearly misguided. We have also reported the possibility of the review being AI-generated to the chairs. To name a few:
> - Weaknesses 3-11 are generic and can be directed towards any evaluation and analysis work published in ICLR, without any discussion on how exactly those undermine the value of our work
> - Weaknesses 4,5 directly contradict strength 4, as our work cannot be simultaneously have  "mathematical and technical rigor" and lack "equation and notation clarity"
> - Weakness 6 is self-contradicting as the definition of "Qualitative Analysis" is exactly "Few Representative Cases"

---

### Official Review · Reviewer_Y3Qi · 2025-11-03

**Soundness:** 3
**Presentation:** 3
**Contribution:** 3
**Rating:** 4
**Confidence:** 3

**Summary:**

The paper introduces two new benchmarks, BLOCKSWORLD-REAL and ALFRED-MULTI, which for the first time establish partially observable, multi-view environments within the Vision-PDDL-Planning framework. It further designs five VLM-as-Formalizer pipelines that generate PDDL problem files defined by the objects, initial states, and goal states of the given environment, and then use a PDDL solver to compute valid plans. Compared with the end-to-end baseline, the proposed VLM-as-Formalizer pipelines demonstrate significantly stronger performance on both benchmarks. Moreover, the paper shows that the primary weakness of current vision-language models lies not in their language reasoning ability but in their visual understanding, as they struggle to accurately capture the relationships between objects within the environment.

**Strengths:**

- The proposed benchmarks bridge the gap between synthetic and real-world embodied environments.

- The paper provides a comprehensive comparison by evaluating five structured VLM-as-Formalizer pipelines, which helps clarify the role of intermediate representations in enhancing visual understanding.

- The evaluation metrics are well designed.

**Weaknesses:**

- All proposed pipelines are primarily prompt-engineering variants rather than algorithmically distinct methods, without introducing fundamentally new learning mechanisms.

- The absence of concrete prompt templates of these baselines somewhat limits reproducibility and interpretability.

**Questions:**

- The generation process of the VLM-as-Planner baseline seems somewhat vague. It is unclear how the model actually produces the plan. Does it first generate a caption or a structured scene description and then predict a sequence of actions defined in the PDDL domain file, or does it directly output the action sequence? Additionally, would introducing intermediate representations, as done in pipelines B–E, help this process? For instance, if the environment is first formalized through a Caption- or Scene-Graph-based representation and then used as input for planning, would this improve its performance?

- Have the authors considered evaluating their pipelines on additional models (e.g., Claude or Gemini) to examine the generality of the findings?

---

> ### Author Response · Authors · 2025-11-18
> **Intent behind a Single VLM-As-Planner Baseline**
>
> While we agree with the reviewer that the paper has a single VLM-as-planner pipeline that can benefit from similar scaling methods, we would like to reaffirm this as a deliberate choice to include only the most representative pipeline as a baseline. In line with previous works that study PDDL pipelines [1, 2], the focus of the paper is to thoroughly evaluate the strengths and weaknesses of VLM-as-formalizer techniques, with VLM-as-planner serving only as a useful reference, particularly used to compare against the Direct-P pipeline with the same inputs.
>
> [1] Wong et al., Learning adaptive planning representations with natural language guidance, 2023 (ICLR 2024)
>
> [2] Huang et al., On the Limit of Language Models as Planning Formalizers, 2025 (ACL 2025)

---

> ### Author Response · Authors · 2025-11-18
> **Inclination against Evaluating Further Close-source Models**
>
> While we agree that additional closed-sourced models apart from the existing GPT-4.1 evaluation could provide useful information, our evaluation intentionally chooses to select an SOTA open-source model (Qwen 2.5 VL) to ensure reproducibility that the reviews themselves pointed out above. This preference towards open-source models is consistent with the reviewing guidelines of some conferences like ACL, which discourages comparisons to closed-sourced models due to methodological concerns.

---

> ### Author Response · Authors · 2025-11-18
> **Novelty of the VLM-As-Formalizer Framework**
>
> While it is possible to label our specific **pipelines** as “prompt-engineering”, we would like to point out that the VLM-as-formalizer **methodology** is a novel and promising concept as we have clearly argued in Section 1. As mentioned by reviewer Q21t, the related works of this methodology are scarce, and as agreed upon by all reviewers, the existing evaluation is insufficient. In past ICLR oral presentations, we consistently observe works that similarly proposed novel methodologies with thorough empirical evaluations despite simple pipelines, such as [1, 2, 3]. We argue that our comprehensive evaluation of practical pipelines is a necessary prerequisite for future work that advances algorithmically and theoretically.
>
> [1] Yue et al., Inference Scaling for Long-Context Retrieval Augmented Generation, 2024 (ICLR 2025 Oral)
>
> [2] Weng et al., Do as We Do, Not as You Think: the Conformity of Large Language Models, 2025 (ICLR 2025 Oral)
>
> [3] Song et al., Mind the Gap: Examining the Self-Improvement Capabilities of Large Language Models, 2025 (ICLR 2025 Oral)

---

> ### Author Response · Authors · 2025-11-18
> **Reproducibility with Released Prompt Templates**
>
> We would like to acknowledge the fact that we did not manage to upload the codebase before the deadline, and have now included it in the supplemental materials.
>
> We also thank the reviewer for requesting prompt templates for reproducibility. We have now attached the complete suite of templates for each of the six pipelines in the newest PDF, which together with our released code allows full reproduction of the results in Section 5.
>
> We also provide the following snippets below:
>
> **Caption-P**
>
> - First step: caption generation.
>
> ```
> You are helping a robotic planning task.
> Given the image of a scene and an instruction, generate a detailed scene description.
>
> For the current domain, this is the domain file:
> {{PDDL_domain_file}}
>
> {{environment_specifics}}
> Instruction: {{instruction}}
>
> Analyze the provided images and generate a detailed scene description that focuses on:
> 1. Objects and their types (especially those defined in the domain above)
> 2. Quantities of each object type
> 3. Spatial relationships between objects (particularly those matching domain predicates)
> 4. Current state of each object relevant to the planning task
> 5. Colors, positions, and other visual properties that help distinguish objects
>
> IMPORTANT:
> - Focus on objects that are relevant to the domain types and the given instruction
> - Describe relationships that match the predicates defined in the domain
> - Be specific about object names and their distinguishing features
> - Only generate a scene description, do NOT generate any PDDL
>
> Provide a comprehensive scene description:
> ```
>
> - Second step: PDDL generation.
>
> ```
> You are helping a robotic planning task.
> Given the image of a scene and an instruction, generate the PDDL file with objects, initial state, and goal specification.
> Your output should adhere to the constraints defined in the domain file.
> You must output the PDDL file in the correct format.
> Examples of PDDL problems given a different domain instruction:
> {{PDDL_problem_file_for_hanoi}}
>
> For the current domain, this is the domain file:
> {{PDDL_domain_file}}
>
>     The image of the scene has been provided.
>     Please first analyze the image and then generate the PDDL problem.
>
> {{environment_specifics}}
> Instruction: {{instruction}}
>
> SCENE CAPTION:
> {{caption}}
>
> Based on the above scene caption, the images, and the instruction, generate the PDDL problem file.
> Use the caption to understand what objects are present and their current state.
> Make sure the PDDL problem accurately reflects the scene described in the caption.
>
> Generate the PDDL problem:
> ```

---

### Author Response · Authors · 2025-11-30
**Summary of all concerns addressed**

We thank the reviewers and AC for their efforts. To help the AC navigate the discussion, we provide a summary below.
- Reviewer Y3Qi's (rating of 4) primary concerns are the novelty and reproducibility. We have cited past ICLR publications that made similar contributions and provided a plan to better outline our novelty. We have also provided the prompt templated as requested to maximize reproducibility. All other reviewer's questions have been clarified.
- Reviewer q21t's (rating of 4) 17 weaknesses and questions are largely tangential and factually wrong. We have made a detailed argument on how this review may be AI-generated and does not consitute a valid judgement of our work.
- Reviewer BYka and m2AE (rating of 6, said would increase score) raised only minor concerns on phrasing and technical details. We have clearly addressed each by providing clarification that will be added to the camera-ready version of the paper.

---

### Meta-Review · Area_Chair_17wR · 2026-01-03

**Summary:**

The paper introduces VLM-as-Formalizer pipelines for one-shot, open-vocabulary, multimodal PDDL formalization. Leaving out one potential LLM-generated review, which I do not take into consideration in the meta-review, there are several concerning points raised in the initial round. The major concern is insufficient novelty. Reviewers seem to recognize and appreciate the contribution of the newly proposed benchmarks and the comprehensive comparisons. However, they still feel that the newly introduced prompt-engineering methods are not sufficient for ICLR publication.

There is also additional request for experiment, including evaluating further models to examine the generality of the findings (Reviewer Y3Qi).

There are criticisms regarding reproducibility and interpretability, mainly due to the missing explicit prompt templates and an unclear description of how the VLM-as-Planner baseline generates plans.

In addition, there is a concern regarding the most accurate pipelines are highly token-inefficient, and the role of the VLM is closer to state formalization than true multi-step planning, which somewhat overstates the claimed “thinking” capability.

**Reviewer Concerns:**

For the concerns and questions regarding reproducibility and interpretability, practicality concerns of the pipelines, etc., the authors provide a detailed response, such as the used prompt templates, which I believe will be satisfactory to the reviewers. However, the major common concern is technical novelty. In the rebuttal, the authors argue that the VLM-as-Formalizer methodology is novel and provide prior ICLR accepted papers. I believe the reviewers would appreciate the rebuttal and recognize that the proposed pipelines are new, but to me this may not justify that this work is sufficient for ICLR.

Additionally, the authors show an inclination against evaluating further closed-source models in the rebuttal to Reviewer Y3Qi. I also believe this question is necessary to respond to. I notice that the two shown results in Table 1 using G-4.1 and Q-72B seem a bit inconsistent. For instance, in G-4.1, SG-P seems better than Caption-P; however, in Q-72B, Caption-P is much better than SG-P. Testing on more models may provide insights on how to choose a better design. Also, Reviewer Y3Qi mentions “additional models,” not necessarily only closed-source models, although they did list Gemini and Claude as examples. So i feel this concern is still outstanding.

**Reviewer Scores:**

I am not considering the negative score from Reviewer q21t due to the potential LLM-generated review. I also notice that Reviewer m2AE has a low confidence score (2) and provided a short review. Reviewer BYka may consider raising their score. Meanwhile, Reviewer Y3Qi may keep the negative score due to the concerns about insufficient novelty and the missing evaluation of other base models not being fully addressed.

---

### Decision · Program_Chairs · 2026-01-26

Reject